# Coastal futures: New framings, many questions, some ways forward

Tom Spencer[1] (ORCID), Janine Adams[2] (ORCID), Martin Le Tissier[3] (ORCID), A. Brad Murray[4] and Kristen Splinter[5] (ORCID)

[1]Department of Geography, University of Cambridge, Cambridge, UK; [2]Institute for Coastal and Marine Research, Nelson Mandela University, Gqeberha, South Africa; [3]Coastal Matters Ltd, Newcastle upon Tyne, UK; [4]Division of Earth and Climate Sciences, Nicholas School of the Environment, Duke University, Durham, NC, USA and [5]Water Research Laboratory, School of Civil and Environmental Engineering, UNSW Sydney, Sydney, NSW, Australia

## Perspective

**Keywords:**
coupled social–ecological systems; nature-based solutions; coastal justice; Anthropocene coasts; adaptive pathways

**Corresponding author:**
Tom Spencer;
Email: ts111@cam.ac.uk

### Abstract

Although coasts are frequently seen as at the frontline of near-future environmental risk, there is more to the understanding of the future of coastal environments than a simple interaction between increasing hazards (particularly related to global sea level rise) and increasing exposure and vulnerability of coastal populations. The environment is both multi-hazard and regionally differentiated, and coastal populations, in what should be seen as a coupled social–ecological–physical system, are both affected by, and themselves modify, the impact of coastal dynamics. As the coupled dance between human decisions and coastal environmental change unfolds over the coming decades, transdisciplinary approaches will be required to come to better decisions on identifying and following sustainable coastal management pathways, including the promotion of innovative restoration activities. Inputs from indigenous knowledge systems and local communities will be particularly important as these stakeholders are crucial actors in the implementation of ecosystem-based mitigation and adaptation strategies.

### Impact statement

Coasts are shaped by both processes of natural change and changes in patterns of use of space and resources by societies. Coasts are particularly sensitive to global change and continuously respond to these changes, often in ways that lead to new threats, or exacerbate existing threats, to the very supply of goods, services and space that make them so attractive to societies. Transformations for sustainable development at the coast present both a considerable set of challenges and opportunities. We argue that understanding the future of coastal environments requires a much wider, and more sophisticated, framing than simply considering the impacts of global sea level rise on rapidly growing coastal populations. In essence, the globalised view needs to be extended into arguments that stress: the importance of place-and people-based biophysical systems and social contexts; the need for technical innovation in restoring and rehabilitating degraded coastal environments; and the identification of just and equitable adaptive response pathways towards better, more robust coastal futures.

### Introduction

Across polar, temperate and tropical regions, coastal zones collectively define a critical interface of the global ocean–atmosphere system. Arguably among the most transformed landscapes on Earth, coasts consist of sensitive social–ecological–physical systems, deeply embedded in many of the UN sustainable development goals (SDGs). While the oceans clearly relate to SDG 14 (life below water), coastal environments also play key roles in the eradication of poverty (SDG1); the ending of hunger and the sustainability of agriculture (SDG2); the provision of affordable and clean energy (SDG 7); the adaptation of coastal cities and communities to environmental change (SDG 11); the good use of resources, including food security (SDG 12); responses to climate change (SDG 13); and the promotion of just and equitable societies, including the maintenance of sovereign boundaries, under ever-changing coastal landscapes (SDG 16). Yet whilst coasts are placed high on the ladder of planetary environmental concerns (e.g., Oppenheimer et al., 2019), this positioning is often framed in terms of the single, coupled interaction of, on the one hand, the hazard of global mean sea level (GMSL) rise and, on the other hand, of the exposure and vulnerability that comes from the rapid growth of coastal populations (both resident and transient from tourism) and accompanying infrastructure. GMSL is projected to rise between 0.38 [range: 0.28–0.55] m (SSP1–1.9) and 0.77 [range: 0.63–1.01] m (SSP5–8.5) by 2,100 relative to 1986–2005 under low and high emissions scenarios, respectively. However, greater GMSL rise could be caused by the earlier-than-projected disintegration of marine ice shelves, the abrupt,

widespread onset of marine ice sheet and ice cliff instability around Antarctica, and faster-than-projected changes in the surface mass balance and discharge from the Greenland Ice Sheet (Fox-Kemper et al., 2021; Slangen et al., 2023). Presently (2023), 2.15 billion people live in the near-coastal zone and 898 million in the low-elevation coastal zone globally. These numbers could increase to 2.9 billion and 1.2billion, respectively over the 21st century, depending on adopted socioeconomic scenarios (Reimann et al., 2023). Of the 37 megacities (populations in excess of 10 M), 62% can be considered 'coastal' (Barragan and de Andres, 2015).

Modern developed coasts constitute a fundamentally new type of system, one with characteristics that result from tight couplings between human and natural coastal dynamics (e.g., McNamara et al., 2023), in places creating totally new, artificial coastal land areas (e.g., Sengupta et al., 2023). The common framing that sees causality running from the physical environment to its social impacts often results in human influence forming the final section of an academic paper or the last chapter in a coastal textbook. But rather than 'natural systems with humans disturbing them' one can tell a more compelling alternative story, one of 'human systems, entwined with natural systems embedded within them'. On some coasts, recent settlers have arrived with little, or no, knowledge of the coastal environment and the potential flood hazards they may face whereas elsewhere there may be deep knowledge from long histories of interaction with their natural surroundings. Climate adaptation strategies need to be fitted to the communities at risk, including, where they can, the incorporation of indigenous knowledge and values (David-Chavez and Gavin, 2018; Proulx et al., 2021). More broadly, there will be a need to move beyond solely technocentric thinking at the coast, including considerations of the role and juxtaposition of the arts and humanities with the natural sciences.

It follows, therefore, that transformations for sustainable development at the coast present both a considerable challenge and a particular opportunity. We argue that understanding the future of coastal environments requires a much wider, and more sophisticated, framing than simply considering the impact of GMSL on a coastal population metric. In essence, the often globalised view needs to be extended into arguments that stress the importance of location-and people-based biophysical systems and social contexts, including the range of possible just and equitable adaptive responses to change (in the broadest sense).

## Coastal dynamics beyond the single metric of global sea-level rise

Quite different from a simple global signal, there are many different regional rates of relative sea-level (RSL) rise; some regions exhibited rates of RSL change five times the global mean between 1993 and 2003 (Bindoff et al., 2007). Inter-annual RSL variability is also likely to be significant, into decimetres in places, in the near future (e.g., Palmer et al., 2020) and it may take decades for the signal of anthropogenically forced RSL change to be clearly detectable (Lyu et al., 2014) and influence shorelines (D'Anna et al., 2022). Micro-tidal coasts are likely to be more vulnerable to global change, with the effects being seen earlier, because the sea level rise signal becomes a larger proportion of the total water level (tide plus sea level rise) signal (Ponte et al., 2019). Furthermore, coastal change is driven not only by these slow onset, chronic dynamics (sea level rise and additionally ocean warming and ocean acidification) but also by acute shocks, from tropical and extratropical storms, associated storm surges, marine heat waves, coastal catchment wildfires, and freshwater flood inputs (e.g., Kirezci et al., 2020; Warrick et al., 2023). All these impacts show high regional variability; thus, for example, while coasts on the western margins of the Pacific basin sit within the tropical cyclone belt and are at risk from storm surges, low-lying islands in more equatorial and eastern Pacific Ocean locations experience flooding from the combination of distant-source swell waves, king tides and sea level rise (e.g., Tuvalu and Kiribati: Hoeke et al., 2021). Recent 'heat domes', catastrophic fluvial flooding, extensive droughts and large-scale coral bleaching events suggest that the climate change signal may well be first seen in these, until now rare, extreme events, with shortening recurrence intervals between them (Seneviratne et al., 2021). And compound extreme events (e.g., storm surge high water levels accompanied by freshwater runoff from heavy rains, or a sequence of tropical hurricanes making landfall) add further complexity to system response (Zscheischler et al., 2018). Concentrating solely on sea level rise drives a focus on a 'future problem' of slow vertical system response and neglects the other processes that shape our coastlines. These include settings where longshore processes, and thus lateral system response, as well as shorter-term processes, such as storms, are often dominant (e.g., for the under-assessment of extreme water levels compared to the focus on sea level rise see Wahl et al., 2017). Furthermore, a sea level rise focus underplays those critical controls on system health that are only weakly linked to climate change, such as changes in fluvial and/or marine sediment supply.

## Coastal systems as integrated biophysical systems

A significant proportion of coastal systems are vegetated (tidal wetlands, sand dunes, coastal cliffs) or are located behind biosedimentary systems (coral reefs, seagrass-floored lagoons) in interconnected coastal 'seascapes' (Ogden et al., 2014). Knowledge about the entrainment, transport and deposition of coastal sediments and the nature of, and controls on, coastal ecology are well established, although research often reflects the ease of access (e.g., locations near field stations and institutions in developed economies, often in the summer months). Furthermore, physical science typically concentrates on monitoring (which prioritises the temporal element) whereas biological science largely focuses on sampling (which favours the spatial element). In addition, understanding the processes that shape and reshape coastal biophysical environments requires addressing the couplings between ecological and physical processes, in contrast to studying them separately (e.g., D'Alpaos, 2011; Durán Vinent and Moore, 2015). On many coasts, therefore, there is an urgent need to merge these different perspectives across disciplines and address coastal dynamics as an integrated biophysical system (Solan et al., 2023). And as satellites open up the possibility of large spatial and temporal scale monitoring of a variety of coastal processes and ecosystem health monitoring, more holistic observational approaches become possible (e.g., Blount et al., 2022; Vitousek et al., 2023). These new theoretical and observational approaches are particularly important given the growing interest in non-structurally engineered responses to coastal erosion and shoreline retreat (Van Zelst et al., 2021; Wedding et al., 2022). Nature-based coastal protection offers the promise of long-term sustainability as, with adequate sediment supply, coasts have the potential to respond to environmental forcing, including the tracking of rising sea levels (Spalding et al., 2014). However, nature-based

approaches to enhancing resilience in the near future can in some cases reduce resilience over decadal timescales. Thus, for example, building artificially high barrier island dunes reduces storm over-wash in the short term but promotes narrower and lower islands in the long term, so that when dunes are overtopped, sediment redistributions – and impacts on island infrastructure – are more immediate and severe (Magliocca et al., 2011). Understanding, and thus aiding, the trajectory of a coastal ecosystem's ability to adapt and maintain functionality is fundamental to the long-term maintenance of natural capital and the delivery of ecosystem services. But what exactly do we know about the design rules and implementation practices to successfully work with natural processes in coastal risk management? (e.g., Orton et al. (2023) on the knowledge gaps behind the construction and operation of gated storm surge barriers). And given that risks and habitability (e.g., Horton et al., 2021), over decadal timescales, change as the landscapes and ecosystems change, what practises will enable long-term, mutual resilience of coastal landscapes, ecosystems, and societies?

## Coasts in the Anthropocene

On developed coasts, physical and ecological processes clearly impact human communities; the very processes that create and maintain coastal environments, including sea-level rise, storms, and ecological change, often represent hazards for people, structures and infrastructure. On the other hand, human decisions and actions have resulted in the removal, degradation and fragmentation of coastal environments and have lessened the ability of these systems to adapt to climate change and act as natural protective barriers for coastal populations (Simkin et al., 2022; De Dominicis et al., 2023). These impacts are not new; there have been centuries of modification, degradation and loss from the anthropogenic activities of land conversion (for agriculture, aquaculture, industry, housing and infrastructure) and misuse (dredging and canalisation, waste disposal and pollution) although one might argue that the scale of human impact – and its consequences – has greatly increased in the last 100 years. Thus, economic damages to coastal assets from tropical cyclones at 2100 are projected to increase by >300% on present values due solely to coastal development, a much larger effect than that projected for climate change impacts, even under high-end climate warming scenarios (Gettelman et al., 2018).

It is clear that to understand how this complex socio-ecological-physical system works at the coast, and how it might evolve over decades as the coupled dance between human decisions and coastal environmental change unfolds, will require transdisciplinary approaches. Deep collaborations among stakeholders, practitioners, and researchers from a wide range of physical, biological and social sciences will be needed. Forging new, transdisciplinary science will produce the understandings that will inform decision-making, allowing communities and societies to evaluate the long-term outcomes of decisions with which they are faced, both now and to mid-century.

Transdisciplinary approaches will also be necessary to inform innovative restoration activities. Anthropogenic pressures are driving coastal ecosystem loss and coastal squeeze. However, restoration research is growing and there are important lessons to be learnt from those activities which analyse the ecological and social effects of restoration activities. A socio-environmental approach addresses the gap between governance, implementation and social

commitment and allows for transfer of knowledge across the science-policy-practice continuum. Innovative approaches for water quality improvement, for hydrological reconnection and for restoring ecosystem services and societal benefits are priority areas for investigation. Inputs from indigenous knowledge systems and local communities are particularly important as these stakeholders are crucial actors in the implementation of ecosystem-based mitigation and adaptation strategies (Porri et al., 2023).

For sustainable coastal futures the management, development, and use of the coast's resources requires critical and reflective research across all disciplines to realise transformations to prevailing practices, institutional structures and processes, including consideration of the inter-connected ethical, cultural, political, social, economic, institutional, technological and behavioural dimensions of coastal development. There are many gaps in our knowledge and understanding about how to transform prevailing coastal thinking and practices. How can formal and informal coastal governance structures and processes be aligned to foster resilience, adaptive capacity and sustainability? What does experience in different coastal contexts reveal about limits, barriers and opportunities? What can be done to support and empower coastal communities that are most at risk in this era of global change – the exposed and vulnerable communities in low-lying deltaic regions (Shaw et al., 2022); sea level atoll islands (Duvat et al., 2021; Cooley et al., 2022); and the circum-arctic region (Ford et al., 2021)? How can conflicting beliefs, worldviews and practices be reconciled? What can be done to align interests and coastal equity and sustainability prospects across sectoral, institutional, geographic and temporal scales? What can be learned from past transformative coastal change? Such questions necessitate reframing how we understand and take actions in pursuit of 'human progress' with vitally important implications for how we reconcile individual, group and societal interests, rights and responsibilities within and between generations.

## Conclusions

Finally, whilst we recognise that the goal of environmental sustainability is laudable in principle, what is it that can be sustained if all around is changing rapidly and continuously towards an unpredictable future? We argue that the goal of achieving sustainability should be replaced by a strategy of eliminating manifestly unsustainable practices. This is challenging when there are considerable legacy issues on many coasts (including, e.g., the decommissioning of ageing nuclear power stations and identifying and remediating coastal landfill sites). But we call on all academic, policy and governance communities to provide the underpinning principles and understandings, both quantitative and qualitative, that will allow societies to properly identify sensible trajectories and thus proactively steer themselves towards better coastal futures.

**Open peer review.** To view the open peer review materials for this article, please visit http://doi.org/10.1017/cft.2023.22.

**Data availability statement.** Data availability is not applicable to this article as no new data were created or analysed in this study.

**Author contribution.** All authors contributed to the conception and design of the paper, initial drafting and subsequent revisions.

**Competing interest.** The authors declare none.

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
