## [Reviewer Report]

Thank you for inviting me to review this Perspective piece by the Editor-in-Chief and Senior Editors. This is a well written piece and I have no major concerns. Some of the key messages that the authors could be stronger so that there is a clearer argument throughout, as it is more a statement of facts, rather than a visionary strength or weakness of a scientific theory or hypothesis. Other aspects to be considered include:

Line 64 – ‘endangered’ I’m not sure whether I would use this word. A landscape (unless biological) is not going to die off!

Line 67-70. Please consider SDG 1 and/or 2 – zero poverty and no hunger. Mega deltas are major food basins affected by salinity and sea-level rise. See https://doi.org/10.1016/j.jenvman.2020.111736

Line 92 – I agree with you here. It also involves legislation and regulations that were targeted for inland areas, or those inland knowing that the coast will affect them through migration or access (tourism).

Line 118 – I agree here with the fact that SLR has been pushed as the main driver of change. Population and development is a lot greater, and has been for a long time. It’s harder to consider due to the human element, The first is not appropriate to cite, but see: https://in.pinterest.com/pin/470766967272221885/ or read here https://doi.org/10.1029/2022EF002927 You could also mention here changing extremes and their growing importance in impact assessments as they have traditionally been overlooked in this area (e.g. https://doi.org/10.1038/ncomms16075). Also how the coast is used as a means of resources / employment in developing nations that are projected to rapidly expand. There are lots of issues here, e.g. sandy mining that could be explored, as this is a key feature of future coastlines.

Line 136 – I agree with this point too on merging on systems, as we all like to think in silos.

Line 178 – This is particularly so where there are legal obligations, such as with coastal squeeze and the resulting loss of habitats.

---

## [Reviewer Report]

This article proposes that the current framing coastal research is too limited in scope to fully embrace the problem of sustainable development in coastal areas, as it often focuses on assessing sea-level rise impacts and stakes at risks, without considering the broader aim of ensuring sustainability in the long term or identifying sustainable coastal management pathways and achieve SDGs in coastal areas. The paper proposes transdisciplinary research to address this. The paper is short, concise and well written, which is excellent. In my view it completes and extends the perspectives ot previous authors in a usefull and timely way, e.g. Brown et al (https://www.nature.com/articles/nclimate2344), and therefore can be an excellent contribution for researchers and practitionners.

I think that this paper is relevant and fully in scope to coastal futures and I have minor comments.

Two general minor comments:

- In the worst case, the perspectives for sustainable development pathway may be very limited and the habitability of coastal zones may be questioned. Examples could be low-lying megadeltas affected by reduced sediments inputs and at risks of cyclones or storms, or small islands. May be this specific case would deserve a note. This question of habitability is assessed in the chapter 15 of the IPCC for small islands https://www.ipcc.ch/report/ar6/wg2/downloads/report/IPCC_AR6_WGII_Chapter15.pdf

- If the authors can include a conceptual figure in their paper, illustrating their key idea (how the new framings and proposed way forwards can address the challenge of sustainable development in coastal areas), this would be potentially usefull.

Page 3 line 76 : I would suggest to add uncertainties around these median values of sea-level projections and eventually mention that larger sea levels outcomes can not be excluded in case of Antarctica ice-sheets collapse. All arguments on these aspects are presented in Fox-Kemper et al 2021, which the author cite.

Page 3 line 89: the argument is excellent but I find the introduction of indigenous knowledge here a bit abrupt. Indigenous knowledge is defined in the IPCC glossary (WGII 2022) as “The understandings, skills and philosophies developed by societies with long histories of interaction with their natural surroundings”. In many cases there are coastal communities who have settled in coastal zones recently, without much knowledge on this coastal area and without actually considering coastal hazards such as flooding or other coastal conservation issues. May be there is a point to be made on coastal communities in general before explaining how this view integrates indigeneous knowledge?

Page 3 line 115: again I fully agree with the argument but I think the role of extreme events will be different depending on the region. For example, in some tropical regions such as eastern Polynesia or Guyana, cyclones are rare or never happen, and sea level rise may actually first materialize in the form of new or more frequent and more intense chronic flooding events (superimposition of high tides and sea level rise, eventually compound with swells or rainfall). E.g. Moftakhari et al. https://doi.org/10.1029/2018WR022828 - may be a bit of nuance here is needed. Note also that this paper can be useful to support the statement that indirect and cascading impacts of sea level rise are potentially very important, yet there is a research gap in this area

Line 145-148: I think that the example of Magliocca et al is well chosen, but the problem is that you need to read the abstract to understand how the reconstruction and maintenance of dunes can compromise resilience in the long term. Just a bit of reformulation is probably sufficient here.

Line 152: again I think it is an excellent point – here a recent perspective by Orton et al. also supporting this statement in the context of the likely increase of storm surges barriers constructions in estuaries worldwide https://agupubs.onlinelibrary.wiley.com/doi/full/10.1029/2022EF002991

Line 192: I suggest considering adding the environmental dimension here (e.g., ecosystem conservation, ecosystem services management…)

Line 211 and 212: eliminating unsustainable strategies is certainly a sound objective, but I am not sure it is sufficient to guarantee sustainability. In my view there is a legacy from past decisions that require additional innovations to move some coastal systems into a sustainable state. For example, coastal nuclear plants, landfills or polluted soils exist already and will need to be managed.

Typos:

- Line 81 – revision mode activated

I hope that this review is useful

Gonéri Le Cozannet, 01/06/2023

---

## [Editor Report]

Both reviewers agree that this perspective piece is well written and clearly within scope of Coastal Futures, although one notes that the overall argument could be strengthened. I support the reviewrs‘ recommendation of publication after minor revisions. Most of the suggestions should be relatively easy to address; I leave it to the authors’ judgement regarding an appropriate balance between including additional content (or nuance) and keeping the overall perspective piece clear and concise. The suggestion to include a conceptual figure may be rather more challenging to address, although I agree with the reviewer that this would further add to the quality and impact of the article.

---

## [Reviewer Report]

Many thanks for answering the reviewers' comments and providing an easy to read response. All but one have been answered satisfactory. In response to your addition text below, please can you add a supporting reference. Thank you. I look forward to seeing your work published.

R2: Page 3 line 89: the argument is excellent but I find the introduction of indigenous knowledge here a bit abrupt. Indigenous knowledge is defined in the IPCC glossary (WGII 2022) as “The understandings, skills and philosophies developed by societies with long histories of interaction with their natural surroundings”. In many cases there are coastal communities who have settled in coastal zones recently, without much knowledge on this coastal area and without actually considering coastal hazards such as flooding or other coastal conservation issues. May be there is a point to be made on coastal communities in general before explaining how this view integrates indigeneous knowledge?

Response: We adjust our text as follows:

‘On some coasts, recent settlers have arrived with little, or no, knowledge of the coastal environment and the potential flood hazards they may face whereas elsewhere there may be deep knowledge from long histories of interaction with their natural surroundings. Climate adaptation strategies need to be fitted to the communities at risk, including, where they can, indigenous knowledge and values. More broadly, there will be a need to move beyond solely technocentric thinking at the coast, including a consideration of the role and juxtaposition of the arts and humanities with the natural sciences.

---

## [Editor Report]

My thanks to the authors for their revisions and clear responses to the previous review comments. One of the reviewers has noted that the revised text around Page 3 Line 89 (on indigenous knowledge and knowledge of hazards within coastal communities) should be supported by one or more references. I believe that this is a reasonable request, and encourage the authors to include at least one supporting reference to the revised text. After this (very) minor revision, I would be happy to accept the paper without requiring additional reviewer comments.